# Early Detection of Jujube Shrinkage Disease by Multi-Source Data on Multi-Task Deep Network

**DOI:** 10.3390/s25216763

**Published:** 2025-11-05

**Authors:** Junzhang Pan, Lei Zhou, Hui Geng, Pengyu Zhang, Fenfen Yan, Mingdeng Shi, Chunjing Si, Junjie Chen

**Affiliations:** 1College of Information Engineering, Tarim University, Alaer 843300, China; 10757241179@stumail.taru.edu.cn (J.P.); 10757242324@stumail.taru.edu.cn (L.Z.); 10757232289@stumail.taru.edu.cn (H.G.); yjunxi@taru.edu.cn (P.Z.); smdeng@taru.edu.cn (M.S.); 2College of Horticulture and Forestry, Tarim University, Alaer 843300, China; yanfening@163.com; 3Key Laboratory of Tarim Oasis Agriculture, Ministry of Education, Tarim University, Alaer 843300, China

**Keywords:** jujube, shrinkage disease, early detection, RGB imaging, hyperspectral, multi-source data fusion, CNN-MLP

## Abstract

In the arid cultivation region of Xinjiang, China, shrinkage disease severely compromises the quality, yield, and market value of jujube. Published research has achieved high accuracy in detecting larger lesions using RGB imaging and hyperspectral imaging (HSI). However, these methods lack sensitivity in detecting early and subtle symptoms of disease. In this study, a multi-source data fusion strategy combining RGB imaging and HSI was proposed for non-destructive and high-precision detection of early-stage jujube shrinkage disease. Firstly, a total of 317 fruits of the ‘Junzao’ cultivar were collected during multiple stages of natural infection, covering early-stage shrinkage disease detection across different growth stages, including both green and mature red fruits. Secondly, morphological features were extracted from RGB images in multiple dimensions, while a three-stage feature selection strategy combining Principal Component Analysis (PCA), the Successive Projections Algorithm (SPA), and the Genetic Algorithm (GA) was implemented to identify four key wavelengths from HSI. Thirdly, a hybrid convolutional neural network-multilayer perceptron (CNN-MLP) architecture was constructed, with dynamic feature weighting employed to achieve effective multimodal fusion and optimize detection performance. Experimental results demonstrated that compared to the MLP and CNN models, the proposed method achieved approximately 8.0% and 5.4% improvements in accuracy and 38.6% and 32.4% improvements in F1 scores, respectively. It offers a robust and scalable solution for early disease detection and postharvest quality assessment in jujube production.

## 1. Introduction

Jujube (*Ziziphus jujuba* Mill.), commonly known as the Chinese date, is a key economic crop in China and has attracted growing attention in global agricultural sectors in recent years. Xinjiang Province, in particular, has emerged as one of the primary production regions for jujube, with continuous increases in planting area and output driven by its unique climate and soil conditions. This industrial development has not only optimized the rural economic structure but also significantly enhanced farmers’ incomes. Shrinkage disease has surged among jujube trees, primarily caused by Alternaria alternata under high-temperature and high-humidity conditions. Symptoms include wrinkling of the fruit skin, lignification of the pulp, and decreased soluble solids content. Severe outbreaks can cause yield losses exceeding 30%, with individual orchards experiencing losses over 50%. The deterioration in Jujube quality not only reduces fresh Jujube profitability but also restricts processing and export opportunities. Therefore, early and accurate detection of shrinkage disease is crucial for the sustainable development of the jujube industry.

In production, the detection methods relying on manual visual detection and chemical analysis of date fruit shrinkage disease are subjective, inefficient, long detection cycle and destructive sampling, which are difficult to meet the demand for rapid, nondestructive and efficient detection in modern agriculture. Artificial intelligence (AI) technologies are increasingly becoming the mainstream solution in the field of intelligent agricultural detection. Machine learning-based analysis of RGB image and HSI enables precise detection of plant pathological characteristics and comprehensive evaluation of fruit physiological conditions.

RGB imaging has been widely utilized in plant disease detection due to its low cost and operational simplicity [1]. Jayapal et al. [2] developed a deep learning model based on RGB images for detecting root rot in ginseng, which performed well under controlled lighting conditions but exhibited limited robustness in field environments. Building on single-modal RGB approaches, Nan et al. [3] proposed a classification method that integrates RGB and depth (RGB-D) data to enhance the accuracy of maize leaf disease detection, though challenges remained in terms of generalization under complex environmental conditions. In an effort to improve model efficiency, Rathore et al. [4] introduced a lightweight CNN architecture, LW17, for rice disease detection using RGB and grayscale images. While achieving a balance between accuracy and computational cost, the model lacked adaptability to diverse symptom types and environmental variability. Despite these advancements, RGB-based single-modal approaches still face notable limitations in early-stage disease identification, comprehensive feature representation, and cross-environment generalization due to their inability to capture biochemical information [5].

HSI has emerged as a cutting-edge nondestructive detection technology with extensive applications across multiple domains [6]. Thomas et al. [7] employed HSI combined with deep learning to detect early-stage disease in tomato leaves, achieving over 90% accuracy even before visible symptoms appeared. Sun et al. [8] further explored the use of 3D-CNN models to process hyperspectral data for soybean anthracnose identification, identifying 733 nm as a critical wavelength, with a classification accuracy of 95.73%. Additionally, Shi et al. [9] successfully implemented HSI for early field detection of potato late blight, achieving robust performance by selecting key wavelengths 492 nm, 519 nm, and 765 nm through dimensionality reduction. To enhance feature discrimination in hyperspectral images, Yu et al. [10] proposed a feedback attention-based dense CNN (FADCNN) that deploys dual spatial–spectral branches and multiscale attention mechanisms, significantly boosting classification precision. Guerri et al. [11] subsequently highlighted that the CNN-based frameworks, although powerful, often struggle with spectral redundancy and tend to overfit when labeled data are scarce. Javidan et al. [12] utilized an MLP model to detect various fungal diseases in tomato plants based on hyperspectral imaging, achieving satisfactory performance across multiple pathogens. Despite these advances, hyperspectral methods are often constrained by high equipment costs, complex data processing requirements, and sensitivity to environmental conditions, which limit their scalability for large-scale agricultural deployment [13].

Multi-source data-fusion technology has demonstrated significant advantages in automatic crop-disease detection, offering a new pathway for identifying jujube shrinkage disease. Compared with single-modal approaches, jointly leveraging RGB and hyperspectral imaging (HSI) supplies complementary information that markedly enhances the extraction of disease-specific features. Zheng et al. [14] developed a hyperspectral–RGB fusion network (HRFN) that couples super-resolution and spectral-correction modules, achieving 98.36% classification accuracy in soybean-seed damage detection. Similarly, Sun et al. [15] integrated RGB and HSI data with a 3D convolutional neural network (3D-CNN) to identify soybean anthracnose, attaining 95.73% accuracy and effectively highlighting critical spectral bands for early diagnosis. Most recently, Zhang et al. [16] introduced JuDifformer, a transformer–diffusion multimodal framework that fuses RGB and hyperspectral cues for early jujube-disease recognition, pushing cross-orchard accuracy above 97% and demonstrating strong robustness to environmental variability. Additionally, Mesa and Chiang proposed a multi-input deep learning model based on MLP and RGB-HSI fusion for banana grading, demonstrating the viability of MLPs in modeling spectral characteristics [17]. However, because MLPs fail to model spatial structure and local morphology in RGB images, they struggle to represent visible disease manifestations—specifically, surface deformation and texture irregularities.

Although previous studies have made significant progress in plant disease detection using either RGB or hyperspectral imaging, they often face critical limitations when applied to early jujube shrinkage disease. RGB-based methods are sensitive to external morphology but lack biochemical information, making them ineffective for subtle or latent infections [18]. Hyperspectral approaches, while powerful in capturing biochemical variations, are hindered by high equipment costs, complex processing, and insufficient robustness across environmental conditions [19,20]. Existing multimodal methods have improved feature representation but still struggle to balance spectral–spatial complementarity or achieve robust generalization under realistic orchard scenarios [21,22]. To overcome these challenges, this study proposes a multi-source fusion framework that not only integrates phenotypic information from RGB images and biochemical signatures from HSI but also introduces a three-stage wavelength selection strategy and a CNN–MLP dual-branch architecture with dynamic feature weighting. Compared with traditional single-modality or fixed-fusion models, our approach achieves more sensitive and precise detection of early-stage shrinkage disease while maintaining strong scalability for field application.

The integration of RGB and hyperspectral imaging enables the simultaneous acquisition of both phenotypic and biochemical characteristics of jujube fruit diseases, achieving complementary feature fusion and providing a solid foundation for the early and accurate identification as well as comprehensive diagnosis of shrinkage disease. To address this, we propose a three-stage framework: synchronized acquisition and preprocessing of paired RGB–HSI data, adaptive fusion of spectral (MLP) and spatial (CNN) features with learnable weights, and final output of disease probability maps and confidence scores for early orchard-scale management. First, in the RGB imaging branch, an optimized dual-stream feature engineering system is developed to extract multi-dimensional features, including color histograms, texture descriptors, spectral features, and wavelet coefficients from the Jujube surface. Second, in the hyperspectral imaging branch, a three-stage wavelength selection strategy is applied by independently performing principal component analysis (PCA), successive projections algorithm (SPA), and genetic algorithm (GA) on the full spectral data. The intersection of selected wavelengths from these methods yields four critical bands at 553 nm, 747 nm, 1070 nm, and 1075 nm, effectively capturing disease-specific biochemical signatures. The effectiveness of these selected wavelengths is further validated through classification experiments, ensuring their robustness for subsequent modeling. Finally, in the decision fusion stage, spatial features processed by a convolutional neural network (CNN) and spectral features processed by a multilayer perceptron (MLP) are integrated via a dynamic weighting mechanism, where the adaptive parameters α and β are learned during training. Experimental observations show that α and β stabilize near 0.5, achieving a balanced and optimal integration of spatial and spectral information.

This study makes the following key contributions to the field of intelligent detection of jujube shrinkage disease: (1) a multi-source data fusion framework is proposed that combines phenotypic features extracted from RGB images with biochemical features obtained from hyperspectral imaging; (2) a dual-branch deep learning architecture is developed, integrating a convolutional neural network (CNN) for spatial feature learning and a multilayer perceptron (MLP) for spectral feature extraction; (3) a dataset consisting of 317 naturally infected ‘Junzao’ jujube fruits across different developmental stages is constructed to support early-stage detection under realistic field conditions.

## 2. Materials and Methods

### 2.1. Overview

The experimental workflow for detecting shrinkage disease in jujube fruits is shown in Figure 1. It integrates RGB and hyperspectral imaging to capture both morphological and biochemical attributes. RGB imaging records external features, surface texture, and color variations. Hyperspectral imaging captures spectral reflectance from 350 to 2500 nm for internal physiological changes. After preprocessing, RGB images yield morphological descriptors and hyperspectral data provide optimized spectral features. A dual-branch neural network is used, with RGB features processed by a CNN and hyperspectral features by an MLP. Dynamic feature fusion with learnable weights balances spatial and spectral contributions. The fused features predict disease presence and distinguish early infected green jujubes from mature red ones, supporting field and postharvest management.

### 2.2. Shrinkage Disease Dataset for Jujubes

The experimental sampling for this study was conducted in jujube orchards located in Aral City, First Division, Xinjiang Production and Construction Corps, China, selecting the Junzao cultivar (*Ziziphus jujuba* Mill. cv. Junzao) as the research subject. To ensure the diversity and representativeness of the dataset, a total of four sampling sessions were conducted on 15 August, 25 August, 5 September, and 15 September 2024. During the sampling process, fruits were selected according to strict physiological criteria, including uniform maturity, regular shape, smooth and unwrinkled skin, absence of mechanical damage, vivid reddish coloration, and no visible signs of pest infestation or disease symptoms. As shown in Figure 2, the experimental sampling sites are located in Alear, Xinjiang, and the figure also presents representative appearances of healthy and diseased jujube fruits at different stages.

In the early stages of infection, slight localized wrinkling appeared on the fruit surface, accompanied by gradual hardening of the flesh and a reduction in internal sugar accumulation. As the infection advanced, lesion areas expanded, skin color deepened, and the pathological changes progressively penetrated into the internal tissues, with the overall severity of the disease showing a positive correlation with the duration of infection. By systematically collecting fruit samples across different stages of disease development, a complete dataset encompassing early to late stages of natural infection was constructed, providing a solid foundation for subsequent feature extraction, model development, and detection research.

#### 2.2.1. Data Acquisition

(1)RGB Imaging

To acquire high-quality images, each jujube fruit sample was placed on a standardized matte-black platform under controlled lighting conditions. Two diffuse white LED panels were positioned symmetrically at a 45° angle relative to the sample to provide uniform illumination and minimize shadows and specular reflections. The ambient lighting was blocked during imaging to avoid external light interference. RGB imaging was conducted using a Canon EOS 200DII digital single-lens reflex (DSLR) camera (Canon Inc., Tokyo, Japan) equipped with an EF-S 18–55 mm standard zoom lens (Canon Inc., Tokyo, Japan). The camera settings were manually configured to optimize image clarity and consistency across all samples. The resolution was set to 6000×4000 pixels, the ISO sensitivity was maintained at 100, the aperture was fixed at f/8.0 to balance depth of field and sharpness, and the exposure time was adjusted automatically based on ambient conditions. The white balance was manually calibrated using a standard gray card prior to imaging to ensure color fidelity. The working distance between the camera lens and the sample was maintained between 300 and 400 mm. A total of 317 RGB images were acquired, each capturing a single jujube fruit with minimal background interference. All images were saved in JPEG format with the highest quality settings and no compression artifacts. No cropping or background removal was performed at this stage, allowing the raw visual information to be preserved for subsequent analysis. This standardized imaging protocol provided a reliable and reproducible dataset for downstream feature extraction and modeling tasks.

(2)Hyperspectral Imaging

Hyperspectral imaging of jujube samples was conducted using an ASD FieldSpec^®^ 4 Hi-Res spectroradiometer (Analytical Spectral Devices, Boulder, CO, USA), covering a continuous spectral range from 350 nm to 2500 nm. The system offers high spectral resolution, achieving approximately 3 nm in the visible to near-infrared (VNIR: 350–1000 nm) region and approximately 8 nm in the short-wave infrared (SWIR1: 1000–1800 nm and SWIR2: 1800–2500 nm) regions, enabling the precise capture of subtle biochemical variations associated with early-stage disease progression. To ensure standardized and reliable spectral measurements, all samples were placed on a matte black background during acquisition, minimizing ambient light interference and background reflectance artifacts. Measurements were conducted indoors under controlled lighting conditions to further enhance data consistency. The fiber-optic probe was held in direct contact with the sample surface during acquisition to maximize signal strength and stability while effectively minimizing the effects of specular reflection.

For each individual fruit, five independent hyperspectral scans were collected sequentially. This multi-scan strategy was implemented to mitigate random measurement noise, account for local surface heterogeneity, and improve overall spectral stability. Instrument acquisition parameters, including integration time and the number of scans averaged, were adaptively optimized for each sample to ensure a high signal-to-noise ratio while preventing detector saturation. After acquisition, all hyperspectral data were saved in ENVI (5.3) format (.hdr + .dat) without any preliminary spectral reduction, ensuring the preservation of full-band information for subsequent preprocessing, feature selection, and model development. A total of 1585 hyperspectral samples (317 fruits × 5 replicates) were thus obtained, establishing a robust basis for downstream shrinkage disease analysis.

For each fruit, RGB imaging and hyperspectral scanning were performed consecutively on the same sample placed on the standardized matte-black platform, so RGB images and hyperspectral measurements were paired at the sample level. We did not perform pixel-wise spatial registration between the RGB images and hyperspectral datacubes; instead, morphological descriptors extracted from RGB images and spectral features from selected HSI bands were fused at the sample/feature level. This sample-level fusion preserves complementary phenotypic and biochemical information while avoiding the complexity of strict pixel-to-pixel alignment.

#### 2.2.2. Data Preprocessing

(1)RGB Feature Extraction

The study presents a standardized framework for preprocessing and feature extraction of RGB images. The proposed methodology integrates three key components: comprehensive color space transformation techniques, robust segmentation combining adaptive thresholding with morphological operations, and precise region-of-interest optimization using semi-automatic GrabCut algorithm. For the processed jujube images, the system effectively extracts multi-dimensional features including color attributes, texture characteristics derived from GLCM and LBP algorithms, as well as frequency-domain features obtained through Gabor filters and Haar wavelets, ultimately establishing a comprehensive feature descriptor system. The overall workflow of RGB image preprocessing is illustrated in Figure 3.

To ensure that each jujube fruit completely filled the image sensor while retaining fine spatial detail, the required focal length f was derived from the geometric relationship among the working distance WD, the sensor width V, and the desired horizontal field of view H. This relationship is expressed in Equation (1). By inserting the fixed working distance (300–400 mm) and the Canon EOS 200D II sensor width (22.3 mm) into Equation (1), a focal length of approximately 35 mm was selected. This value guarantees that the entire fruit occupies the frame without cropping while preserving high spatial resolution for subsequent feature extraction.(1)$$f=\frac{WD\times V}{H}$$
where f denotes the focal length, WD represents the working distance, V is the camera sensor width, and H refers to the horizontal field of view. This calculation ensured that the entire fruit body was captured within the frame while maintaining high spatial resolution.

HSV Color Space Transformation and Adaptive Segmentation. To enhance robustness in color analysis under varying illumination conditions, all RGB images in this study were converted to the HSV (Hue, Saturation, Value) color space. The HSV transformation effectively separates chromatic and luminance components, thereby increasing sensitivity to subtle color variations associated with early-stage disease. The value (V) component is calculated as shown in Equation (2), which provides a standardized basis for subsequent color and texture feature extraction and enables more sensitive capture of phenotypic discoloration in jujube skin. In the initial threshold segmentation stage, adaptive thresholding is applied in the HSV space to dynamically adjust pixel-wise thresholds based on local image statistics, thus addressing the challenge of non-uniform illumination. Following thresholding, morphological closing with a 3 × 3 structuring element is performed to remove noise and fill small holes, ensuring the continuity and integrity of binary mask regions. In this study, adaptive thresholding was implemented using a 31 × 31 pixel local window, with the thresholding function defined in Equation (3), where μ and σ denote local mean and standard deviation, R = 128, and k = 0.34. This finely tuned, dynamic segmentation approach enables precise discrimination of diseased fruit regions from the background, providing a robust foundation for automated detection in complex environments.(2)V=max(R,G,B)(3)T(x,y)=μ(x,y)×[1+k⋅(σ(x,y)/R−1)]

GrabCut Semi-Automatic Segmentation. GrabCut segmentation was initiated by enclosing the target fruit within a rectangular window, after which the binary mask was refined through five iterative updates that simultaneously optimized a colour-based Gaussian-mixture model and a spatial-smoothness constraint. The associated energy functional is expressed in Equation (4), where $L_{p}$ denotes the foreground or background label of pixel p and represents $z_{p}$ its RGB colour vector. The first term quantifies the negative log-likelihood of each pixel’s colour under its corresponding Gaussian mixture, thereby assessing data fidelity, whereas the second term penalises label inconsistencies between four-connected neighbouring pixels, with the penalty weighted by the colour-contrast factor $e^{-\beta \|z_{p}-z_{q}{\|}^{2}}$. The regularisation coefficient λ balances the contributions of data fidelity and boundary smoothness, while β is adaptively determined from the global colour variance of the image. By explicitly coupling appearance likelihoods with contrast-sensitive boundary regularisation, this formulation effectively suppresses background artefacts and preserves fine fruit contours, yielding highly reliable regions of interest for subsequent feature extraction.(4)$$E(L,\theta )=\underset{p\in \Omega }{\sum }-ln(\sum_{m=1}^{K}\pi_{L_{p}m}N(z_{p}|\mu_{L_{p}m},\Sigma_{L_{p}m})+\lambda \underset{\langle p,q\rangle \in N}{\sum }[L_{p}\neq L_{q}]e^{-\beta \|z_{p}-z_{q}{\|}^{2}}$$

Color Feature Extraction. Color features represent both global and local color distributions in an image and are crucial indicators for early-stage color change recognition. In this study, histograms and first- and second-order statistics (mean, standard deviation, skewness, kurtosis) were extracted from RGB and HSV channels, with the mean defined as in Equation (5). The extracted color features were used for multimodal comparison with hyperspectral reflectance features. Unlike previous works focusing on single-channel features, this study’s multichannel and multiscale color statistics provide a rich information foundation for model fusion.(5)$$\mu =\frac{1}{N}\sum_{i=1}^{N}x_{i}$$

Texture Feature Extraction. Texture features describe the spatial distribution of gray values and surface structure attributes in an image. In this study, gray-level co-occurrence matrix (GLCM) statistics were computed in four directions (0°, 45°, 90°, and 135°), and local binary pattern (LBP) histograms were extracted to enhance the description of fine-scale structural patterns. A representative GLCM indicator, contrast, is calculated as in Equation (6). These texture features effectively characterize structural deformations and roughness variations caused by disease. By combining multidirectional GLCM and multiscale LBP, this study enhances the model’s ability to capture microstructural changes in jujube fruits associated with shrinkage disease.(6)$$Contrast=\sum_{ij}\left(i-j{)}^{2}\cdot P(i,j\right)$$

Frequency-Domain Feature Extraction. Frequency-domain features can recognize surface wrinkling and periodic detail variations, aiding in the modeling of complex, multiscale lesion patterns. In this study, 12 Gabor filters (four directions × three scales) and a three-level Haar wavelet decomposition were employed for multiscale feature modeling. The Gabor kernel is defined as in Equation (7). These frequency-domain features enhance the spatial-frequency representation of the image, which is particularly suitable for detecting and classifying lesion regions. The extensive frequency-domain features employed in this study enable effective characterization of structural heterogeneity in jujube fruits at varying disease severities.(7)$$g(x,y)=\exp(-\frac{x^{\prime 2}+\gamma^{2}y^{\prime 2}}{2\sigma^{2}})\cos(2\pi \frac{x^{\prime }}{\lambda }+\phi )$$

(2)Hyperspectral Data Wavelength Selection

This study established a hyperspectral data preprocessing and characteristic wavelength selection workflow to extract the biochemical information related to early-stage fruit shrinkage disease from hyperspectral datasets. The process encompasses systematic spectral calibration, advanced denoising and normalization strategies, and rigorous multi-stage key wavelength selection. By integrating these steps, the proposed approach effectively minimizes noise and redundancy, highlights the most informative spectral signatures, and provides high-quality input features for downstream modeling. This unified strategy ensures that the hyperspectral modality is leveraged to its full potential, laying a solid foundation for the accurate and robust classification of shrinkage disease in jujube fruits.

Spectral Calibration and Standardization. Before downstream preprocessing, the raw hyperspectral radiance data were converted to absolute reflectance by means of a two-point calibration. A Spectralon panel with a certified reflectance greater than 99% served as the white reference, whereas a dark reading captured the detector baseline. The calibrated reflectance spectrum R was computed by Equation (8).(8)$$R=\frac{R_{0}-B}{W-B}$$
where $R_{0}$ denotes the raw spectral signal, B is the dark reference, and W is the white reference. This strict calibration protocol ensured the comparability of spectral data across batches and time points, thus providing a solid and unbiased foundation for the subsequent machine learning analysis of shrinkage disease in this study.

Spectral Denoising and Smoothing. To eliminate high-frequency noise and baseline drift commonly present in hyperspectral signals, the Savitzky–Golay filter was applied in this study, which effectively preserved spectral features while removing random noise. The smoothing calculation is given in Equation (9), where $y_{i}$ is the smoothed value, $x_{i+j}$ is the original spectrum within the window, and $C_{j}$ are the filter coefficients. By selecting appropriate parameters, this study maximally reduced environmental variability and sample heterogeneity, ensuring that the extracted features reflect true biochemical differences.(9)$$y_{i}=\sum_{j=-k}^{k}C_{j}\cdot x_{i}+j$$

Key Wavelength Selection. To minimise spectral redundancy while retaining diagnostically rich information, three classical band-selection techniques—principal-component analysis (PCA), successive projections algorithm (SPA) and genetic algorithm (GA)—were applied independently to the full hyperspectral dataset. Each method produced a ranked subset of informative wavelengths; their intersection yielded four consensus bands at 553 nm, 747 nm, 1070 nm and 1075 nm. This consensus strategy couples the global variance emphasis of PCA, the low-collinearity property of SPA and the classification-oriented search of GA, thereby balancing information richness and compactness. Subsequent ANOVA F-score analysis confirmed the discriminative power of these bands (F > 200 for all four), establishing a concise yet biochemically relevant input space for downstream modeling and substantially improving both interpretability and computational efficiency.

Feature Extraction and Standardization. After wavelength selection, multidimensional spectral features mean reflectance, area under the curve, first and second derivatives, and higher-order statistics were extracted to comprehensively characterize disease-related biochemical changes. All features were normalized using min–max scaling as defined in Equation (10), ensuring uniformity prior to model training. This systematic feature engineering pipeline effectively bridges raw hyperspectral data and machine learning models, significantly improving the discriminative power for intelligent detection of shrinkage disease.(10)$$x_{i}^{norm}=\frac{x_{i}-\min(x)}{\max(x)-\min(x)}$$

#### 2.2.3. Dataset Characteristics

A total of 317 jujube fruits were collected under field conditions, covering the full progression of naturally infected shrinkage disease. Of these, 250 fruits were infected, ranging from early green stage with initial surface depressions to late red stage with pronounced shrinkage, while 67 fruits remained healthy and served as controls. This stratified collection captures the key morphological and physiological transitions associated with disease development and provides a solid foundation for subsequent model training and evaluation. To obtain an unbiased performance estimate, the dataset was randomly divided into training, validation, and test subsets in an 8:1:1 ratio (Table 1). Stratified sampling preserved the stage distribution across all subsets, and the random seed was fixed at 42 to facilitate reproducibility. In total, 317 individual ‘Junzao’ fruits were collected in this study. Each fruit was scanned five times by the hyperspectral instrument to mitigate local measurement noise and surface heterogeneity, yielding 1585 hyperspectral samples (317 × 5). The infected fruits were labeled by disease progression as 120 early-stage, 120 mid-stage, and 77 late-stage samples. For model training we used a stratified split (see Table 1): the training set contains 257 RGB images/1285 HSI scans, with the remaining images/scans allocated to the validation and test sets.

### 2.3. Multimodal Feature Fusion Strategy

This study proposes a multimodal fusion model architecture consisting of a CNN branch and a MLP branch, which are designed to extract deep information from RGB images and key hyperspectral bands, respectively. The two branches are adaptively integrated via a learnable dynamic weighting mechanism, enabling accurate and robust detection of early-stage fruit shrinkage. The overall architecture of the proposed dual-branch multimodal network is shown in Figure 4. The network consists of a CNN branch for spatial feature extraction from RGB images and an MLP branch for spectral feature extraction from hyperspectral vectors. Both feature vectors are dynamically fused via learnable weights (α, β) and fed into a fully connected layer and Softmax classifier for final multi-class fruit shrinkage detection. The detailed design of each module is described in the following subsections.

#### 2.3.1. CNN-Based Feature Learning

The CNN branch is designed to model the spatial morphological features of jujube fruits. The core idea is to leverage multiple convolutional and pooling layers to extract multi-scale features from the input RGB image $I_{RGB}$. External morphological characteristics, including surface texture, wrinkling patterns and colour variations. Pooling layers perform down-sampling and noise suppression, enhancing feature robustness. As the network deepens, local information is progressively aggregated into more abstract and global morphological representations. The resulting feature maps are flattened and passed through a fully connected layer (*FC*) to generate the spatial feature vector $f_{CNN}$ for subsequent fusion. The CNN branch is engineered to extract salient spatial morphological features, such as surface texture, wrinkling patterns, and color variations, from input RGB images resized to 128 × 128 pixels. The architecture begins with a convolutional layer comprising 32 filters (3 × 3 kernel, stride 1), which is followed by a 2 × 2 max-pooling layer for down sampling. Subsequently, a second convolutional block, consisting of a layer with 64 filters (3 × 3) and another 2 × 2 max-pooling layer, is applied to capture more complex hierarchical features. All convolutional layers utilize ‘same’ padding to preserve spatial dimensions, a ReLU activation function to introduce non-linearity, and He normal initialization. The resulting feature maps are then flattened and fed into a dense, fully connected layer of 128 neurons, also with ReLU activation. To mitigate overfitting, a dropout layer with a rate of 0.3 is applied. This process yields a final 128-dimensional spatial feature vector, which serves as a comprehensive representation for the subsequent multimodal fusion stage. The entire branch is trained end-to-end using the Adam optimizer, configured with a learning rate of 0.001, β1 = 0.9, β2 = 0.999, and ϵ = 10^−7^. The calculation process is expressed as Equation (11).(11)$$f_{CNN}=FC(Flatten(MaxPool(Conv(I_{RGB})))$$

#### 2.3.2. MLP-Based Feature Learning

The MLP branch focuses on modeling the key biochemical features of the fruit by processing the selected hyperspectral band vector $x_{HSI}$. The MLP is a classic feedforward neural network consisting of several fully connected layers (FC) and nonlinear activation functions. The four-dimensional hyperspectral vector is first projected through a fully connected layer with activation, followed by Dropout to prevent overfitting, and then through another fully connected layer to obtain a discriminative spectral feature vector $f_{MLP}$. This vector captures physiological changes moisture loss and chlorophyll degradation associated with early-stage shrinkage. The calculation process is as Equation (12).(12)fMLP=W2⋅ReLU(W1xHSI+b1)+b2

Here, $x_{HSI}$ is the input hyperspectral feature vector, $W_{1}$, $W_{2}$ and $b_{1}$, $b_{2}$ are the weights and biases of each layer, ReLU(⋅) is the activation function, and $f_{MLP}$ is the final output spectral feature vector.

#### 2.3.3. Dynamic Multimodal Feature Fusion

To fully leverage the complementary advantages of spatial morphological information and biochemical spectral information, a dynamic fusion mechanism based on learnable weights is employed. Specifically, the spatial feature $f_{CNN}$ output by the CNN branch and the spectral feature $f_{MLP}$ output by the MLP branch are concatenated using the ⊕ operation and are adaptively weighted by learnable parameters α and β. This enables the model to dynamically balance the contribution of each modality during the decision process. The fusion process is expressed in Equation (13).(13)$$F_{\mathrm{fused}}=\alpha \cdot f_{CNN}⊕\beta \cdot f_{MLP}$$

The fused multimodal feature vector $F_{\mathrm{fused}}$ is then fed into a fully connected layer and a Softmax classifier to achieve final multi-class intelligent detection of fruit shrinkage. The calculation process is shown in Equation (14).(14)$$\overset{\wedge }{y}=Soft\max(W_{f}F_{fused}+b_{f})$$

Here, $W_{f}$ and $b_{f}$ represent the weights and bias of the classification layer, respectively, and $\overset{\wedge }{y}$ denotes the probability distribution over all output classes. Equation (13) realizes the dynamic weighted fusion of spatial and spectral features, while Equation (14) completes the final classification decision based on the fused multimodal features.

### 2.4. Performance Evaluation

To thoroughly assess the performance of the proposed model in detecting jujube fruit shrinkage disease, this study adopts several widely used classification metrics, including accuracy (ACC), precision (P), recall (R), and F1-score (F1). These metrics enable detailed comparisons between the model’s predictions and true labels, providing a comprehensive evaluation of model effectiveness across multiple dimensions. The confusion matrix serves as the basis for these calculations. In addition, the receiver operating characteristic curve (ROC) and the area under the ROC curve (AUC) are used to further evaluate the discriminative ability of the classification models. The specific calculation formulas are as follows:(15)$$P=\frac{TP}{TP+FP},R=\frac{TP}{TP+FN},F1=\frac{2P\times R}{P+R},ACC=\frac{TP+TN}{TP+TN+FP+FN}$$
where TP (true positive) is the number of positive samples correctly identified, FP (false positive) is the number of negative samples incorrectly identified as positive, FN (false negative) is the number of positive samples incorrectly identified as negative, and TN (true negative) is the number of negative samples correctly identified. The ROC−AUC metric reflects the model’s ability to distinguish between positive and negative samples, with values closer to 1 indicating superior performance regardless of class distribution.

## 3. Results

To evaluate the proposed model’s effectiveness in detecting jujube fruit shrinkage disease, we compared it against several state-of-the-art deep learning and traditional machine learning models, including MLP, CNN, SVM, RF, KNN, GBM, and a randomly initialized CNN+MLP model. In addition, ablation experiments were designed to further investigate the contributions of different components within the proposed architecture. All models were trained and evaluated under the same dataset and experimental conditions to ensure fairness and reproducibility of the results. In model training, we used several quantitative metrics to assess performance, including overall accuracy (OA), F1 score, precision, recall, and the area under the receiver operating characteristic curve (ROC-AUC). In wavelength selection, we used F1 score analysis to evaluate the most discriminative bands.

The training was implemented in Python 3.9.13 using PyTorch 1.12.1 and executed on a Windows 10 Pro system with an Intel i5-12600KF CPU, 32 GB RAM, and an NVIDIA RTX 4060 Ti GPU (8 GB, CUDA 11.3). Both models used the Adam optimizer (learning rate = 0.001, β_1_ = 0.9, β_2_ = 0.999, ε = 1 × 10^−7^, decay = 0) and binary cross-entropy loss, with a uniform dropout rate of 0.3. Input features were standardized using StandardScaler, and the dataset was split 8:2 with a fixed random seed. A grid search was applied to optimize fusion weight (α) and threshold, selecting the configuration with the highest F1 score.

### 3.1. Model Training

As can be seen from Figure 5, the proposed method (Our) shows a superior performance in terms of convergence speed and final training loss compared to CNN, MLP, and CNN+MLP Random models. Specifically, the CNN+MLP Random model displays the slowest convergence and highest final loss value, stabilizing around 0.3 by the tenth epoch. The CNN and MLP models perform similarly, with rapid loss reductions observed within the first few epochs, eventually stabilizing close to 0.05 and 0.1, respectively. However, the proposed model demonstrates the fastest convergence rate, reaching a near-zero loss value by the fourth epoch and maintaining the lowest loss throughout subsequent training epochs. These results indicate that the proposed method significantly enhances the model’s training efficiency and accuracy compared to other baseline models.

### 3.2. Feature Band Selection

#### 3.2.1. Hyperspectral Feature Band Analysis

A spectral analysis was conducted to compare the reflectance characteristics of different types of jujubes. The hyperspectral values were averaged and normalized for better comparison. Figure 6a illustrates the normalized spectral reflectance of four different grades of shriveled red dates, while Figure 6b presents the spectral reflectance of healthy red dates. As observed in Figure 6a, the reflectance spectra of shriveled red dates show significant variations, with multiple peaks and troughs, indicating differences in surface characteristics and internal composition among different grades. In contrast, Figure 6b shows that the spectral curves of healthy red dates are more consistent, suggesting a relatively uniform reflectance pattern. These spectral differences between shriveled and healthy red dates are primarily concentrated within specific wavelength ranges, indicating that these bands may serve as key indicators for classification.

#### 3.2.2. Spectral Feature Selection

Selecting effective spectral bands is essential for ensuring classification accuracy in jujube fruit shrinkage disease detection, as hyperspectral images contain a large number of redundant and highly correlated bands. We employed three classical spectral bands selection methods: Genetic Algorithm (GA), Principal Component Analysis (PCA), and Successive Projections Algorithm (SPA) to extract the most relevant wavelengths for disease identification. The specific wavelengths selected by each method are summarized in Table 2. The comparative analysis showed that 553 nm, 747 nm, 1070 nm, and 1075 nm were commonly selected by all three methods and have high potential for disease identification. The consensus-based selection strategy identified 553 nm, 747 nm, 1070 nm, and 1075 nm as the most diagnostically significant wavelengths for detecting jujube shrinkage disease, with each band corresponding to specific pathophysiological changes in the fruit. The 553 nm band, located in the green region of the visible spectrum, is highly sensitive to variations in pigment concentration. A decline in reflectance at this wavelength typically corresponds to the degradation of chlorophyll, which is a primary indicator of physiological stress and an early symptom of pathogenic infection [23]. The 747 nm band is situated in the critical red-edge region, an area where spectral reflectance is strongly influenced by both chlorophyll content and the internal cellular structure of the fruit tissue. Changes in this band are therefore powerful indicators of the initial pathological alterations that occur as the disease disrupts cellular integrity [24]. Furthermore, the adjacent bands at 1070 nm and 1075 nm fall within the near-infrared (NIR) spectrum, a region primarily associated with moisture content and the vibrational absorption of O-H bonds in water molecules. These wavelengths are particularly effective at detecting changes in tissue water content and the lignification of the pulp, which are characteristic symptoms of advancing shrinkage disease [25,26]. Collectively, this set of four wavelengths provides a comprehensive spectral signature of the disease, capturing a cascade of biochemical and physical changes from early-stage chlorophyll degradation to later-stage water loss and tissue collapse. To validate the effectiveness of the selected wavelengths in identifying jujube fruit shrinkage disease, we conducted an ANOVA F-score analysis. Wavelengths with F-scores above 200 were considered highly discriminative. The results showed that 553 nm and 747 nm achieved F-scores of 719.05 and 318.56, while 1070 nm and 1075 nm reached 275.55 and 275.84, confirming their importance as key spectral features.

### 3.3. Fusion Weight Selection

To investigate the dynamic contribution of each modality throughout the training process, a detailed analysis was conducted with the results presented in Figure 7. The figure illustrates the optimal fusion weight for the hyperspectral (HSI) modality, alpha, determined at 10-epoch intervals, revealing a dynamic interplay where the instantaneous optimal weight fluctuates as the models learn and refine their feature representations. While the optimal weight at any single training stage can vary, a key observation is that the value frequently oscillates around the 0.5 mark, with the average across all tested epochs being 0.44. Arguing that choosing a weight from a single, specific epoch (e.g., the value of 0.0 at epoch 100) could result in a model that is not generalizable, a weight of 0.5 was ultimately selected for the final model. This choice represents a robust and balanced strategy, ensuring that both the spatial features from RGB images and the spectral features from HSI data contribute significantly to the final classification. This approach prevents the model from becoming overly dependent on a single data source, thereby enhancing its stability and generalizability for unseen data.

### 3.4. Detection of Fruit Shrinking Disease in Jujubes

#### 3.4.1. Quantitative Evaluation

Shrink fruit disease was identified by extracting descriptive color, texture, and spectral features from both RGB and hyperspectral images. This paper compares our model against existing representative methods, including MLP, CNN, SVM, RF, KNN, GBM, CNN-MLP random. Table 3 shows that the strongest overall performance was exhibited by our model, which consistently achieved high accuracy, balanced classification, and robust discriminative power. The CNN model’s accuracy was lower than that of ours by approximately 5%, while its F1 score was lower by about 32%. Traditional machine learning methods, including SVM, RF, and GBM, yielded less balanced outcomes. The CNN-MLP Random configuration produced inferior results, with its accuracy approximately 10% lower and its F1 score about 31% lower than those of our model. These differences show that integrating convolution-based feature extraction with fully connected layers enables the model to capture local and global patterns, yielding a robust and discriminative representation optimal for shrink fruit disease detection. Consequently, our model was selected for its ability to effectively unify convolution-based feature extraction with high-level representation learning. The model effectively captures both local and global patterns in color, texture, and spectral data. Compared to single-model approaches, this integrated design has been shown to significantly enhance classification accuracy and robustness, thereby rendering it an optimal solution for shrink fruit disease detection.

To further evaluate the discriminative performance of different models, the receiver operating characteristic (ROC) curves of all models were plotted, as shown in Figure 8. The proposed model achieved the highest AUC value of 0.979, outperforming traditional machine learning approaches such as Random Forest (AUC = 0.955) and KNN (AUC = 0.935). Although some models, including RF and KNN, obtained relatively high accuracy, their F1 scores were considerably lower, which can be attributed to the class imbalance in the dataset. This imbalance leads to a bias toward the majority class, resulting in inflated accuracy but degraded precision–recall balance. Overall, the ROC analysis confirms that the proposed model exhibits better robustness and generalization capability across different classes.

#### 3.4.2. Qualitative Evaluation

Figure 9a presents the scatter plot of the actual data distribution, where blue and red points represent healthy and shrink-diseased jujube fruits, respectively. Figure 9b shows the scatter plot of the predicted results obtained from the classification model. The comparison between the two plots demonstrates the model’s ability to differentiate between the two categories. The close alignment between the predicted and actual distributions indicates the effectiveness of the model in classifying jujube fruit shrinkage disease. 93.7% of the samples were correctly classified. Some misclassifications are observed, which are likely caused by samples with highly similar features that are inherently difficult to distinguish.

### 3.5. Ablation Studies

#### 3.5.1. Ablation on Detective Components

Ablation experiments were conducted by removing or replacing core modules to validate the effectiveness of our design, as detailed in Table 4. The CNN+KNN model, which replaces the MLP component in our architecture with a KNN classifier, achieved an F1 score approximately 12% lower than that of our model, indicating that KNN is less effective in capturing high-level feature representations. Similarly, the CNN+SVM model, which substitutes the MLP with an SVM classifier, showed an accuracy decrease of about 13% compared to our model. In addition, the CNN+MLP Random model, which shares the same structure as ours but uses random initialization, also performed worse, confirming the critical role of proper weight initialization in achieving optimal performance.

#### 3.5.2. Ablation on RGB Feature Factors

To evaluate the contribution of handcrafted RGB features, we conducted an ablation study comparing the performance of the full handcrafted feature set against models with individual feature groups removed (color, texture, Gabor, and wavelet). As shown in Figure 10, removing any single feature group led to a decrease in model performance, with the most significant degradation observed when Gabor features were excluded (Accuracy = 0.778, F1 = 0.000, AUC = 0.500), indicating that Gabor-based texture information plays a crucial role in distinguishing subtle surface deformation patterns related to early-stage shrinkage. In contrast, excluding color or wavelet features caused only minor fluctuations, suggesting some redundancy between these descriptors. Compared with the CNN trained on raw RGB images (Accuracy = 0.905, F1 = 0.769, AUC = 0.910), the handcrafted feature–based MLP achieved higher accuracy and stability (Accuracy = 0.937, F1 = 0.846, AUC = 0.974), confirming the effectiveness of the engineered RGB features in enhancing discriminative power under limited dataset conditions.

## 4. Discussion

### 4.1. Quality Evaluation of Jujubewa

Hyperspectral imaging (HSI) and RGB imaging have become increasingly prominent in jujube quality assessment due to their ability to capture both biochemical and morphological features. HSI provides rich spectral information across hundreds of continuous wavelengths, which enables the precise evaluation of internal quality parameters including moisture content, soluble solids, and disease-related indicators. Lu et al. [27] reported a classification accuracy exceeding 92% in monitoring the quality of dried jujubes using HSI combined with deep learning algorithms. However, the inherent trade-off between spectral and spatial resolution in traditional HSI systems means they lack the fidelity required to capture surface-level characteristics and therefore fail to detect fine defects on small jujube fruits.

To address this limitation, researchers have investigated multi-modal image fusion strategies that integrate HSI with high-resolution RGB data. This approach enhances feature representation by combining spectral signatures with spatial information on texture and color. Zhang et al. [28] developed a dual-branch deep learning framework that fuses RGB and HSI data, achieving a classification accuracy of 93.7% for detecting jujube shrinkage disease. Jiang et al. [29] further improved detection precision through a modified YOLOv8 network, enabling real-time localization of surface defects in jujube fruit. Moreover, state-of-the-art models RJ-TinyViT and MLG-YOLO have achieved mean average precision (mAP) values exceeding 90% in defect classification and fruit localization tasks [30,31].

In addition to quality grading, HSI has been successfully employed in disease detection. Wang et al. [32] utilized hyperspectral reflectance data to detect black spot disease in winter jujubes during postharvest storage, achieving classification accuracies above 95%. Furthermore, Chen et al. [33] employed unmanned-aerial-vehicle multispectral imaging at orchard scale and observed strong correlations between spectral reflectance indices and internal quality parameters, including soluble solids and moisture content.

This study focuses on the detection of jujube fruit shrinkage disease and proposes a multi-source data fusion framework that integrates hyperspectral and RGB imaging. A dual-branch deep learning architecture based on CNN and MLP is employed to extract and dynamically fuse complementary spectral and visual features. Experimental results demonstrate that the proposed method significantly improves the accuracy and robustness of disease classification, offering a reliable and effective solution for intelligent, non-destructive, and rapid detection of jujube diseases.

### 4.2. Hyperspectral and RGB Imaging Quality Detection in Other Fruits

The same methods have also been successfully applied to the quality evaluation of other fruit species. In mangoes, HSI has demonstrated high effectiveness in estimating soluble solids content and detecting surface defects, with classification models achieving high accuracy through optimized wavelength selection and image segmentation algorithms [34,35]. In pears, machine learning models trained on HSI data achieved precise classification across multiple cultivars, demonstrating the strength of spectral information in internal quality assessment [36]. For strawberries, HSI has been employed to estimate nutrient concentrations in leaves, flowers, and fruits, as well as to identify early-stage bruising damage during postharvest handling [37,38]. In peaches and white grapes, HSI was utilized to monitor both internal and external quality attributes, including firmness, sugar content, and surface morphology [39,40].

In citrus fruits, the integration of HSI with AI-based classification models successfully identified fungal diseases citrus black spot (CBS), providing a reliable non-destructive alternative to manual inspection [41]. In bananas and apples, hyperspectral reflectance data enabled the prediction of nutrient content and geographical origin, while the combination of HSI with Vis-NIR spectroscopy in apples achieved high accuracy in estimating soluble solids content and ripeness [17,42].

These results further support the growing application of multi-source data fusion, where integrating HSI with high-resolution RGB imaging enhances comprehensive quality characterization across multiple fruit species. The synergy of spectral and spatial data provides a robust, non-invasive tool for intelligent fruit grading and disease monitoring, particularly within the domains of postharvest management and precision agriculture.

### 4.3. Failure Samples Analysis

Although the proposed multi-source fusion model demonstrated strong overall performance in the classification of jujube fruit shrinkage disease, several misclassified instances were identified during the evaluation process. To investigate the potential causes of these errors, two representative samples were selected for detailed analysis, as illustrated in Figure 11.

As shown in Figure 11a, the first misclassified sample was a healthy fruit that the model incorrectly identified as diseased. This sample exhibited minor surface bruising and slight discoloration caused by mechanical damage. However, these superficial features closely resembled early symptoms of shrinkage disease in the RGB image, making it difficult for the model to differentiate between the two. In addition, the hyperspectral reflectance curve of this sample showed a noticeable depression in the range from 750 to 1200 nanometers, which is a spectral signature typically associated with early-stage infection. The primary cause of this false positive prediction lies in the strong visual and spectral similarity between mechanical injury and disease-related surface lesions. To ensure annotation quality, all samples were reviewed on-site by experienced agronomists; only a small number of samples were identified as mechanical/handling damage. Figure 11b shows the second misclassified sample, which was an infected fruit that the model incorrectly classified as healthy. The RGB image did not exhibit any obvious shrinkage or surface speckling, and the corresponding hyperspectral reflectance curve lacked the characteristic decline typically associated with diseased samples. However, during physical inspection, the fruit exhibited slight softening, which was not visually apparent and could only be perceived through tactile sensation. This false negative result is primarily attributed to the early stage of infection, where the disease had not yet produced noticeable visual or spectral changes, thus making detection by the model particularly challenging.

Limitation of dataset diversity: We note that the present dataset was acquired from a single cultivar (‘Junzao’) within a single growing region and season. Consequently, while the model shows strong within-dataset performance for early-stage shrinkage detection, its cross-cultivar and cross-region generalization remains to be validated. Users should interpret the current results as evidence of feasibility and within-site robustness rather than as proof of universal generalization.

In summary, these misclassification cases highlight the limitations of the current model in detecting ambiguous or low-contrast symptoms, especially when the disease is at a very early stage. Future improvements should explore attention-based mechanisms and multi-modal sensing approaches to improve the accuracy and robustness of early disease identification.

## 5. Conclusions

This study proposed a multi-source information fusion framework integrating RGB and hyperspectral imaging for early detection of jujube shrinkage disease. The dual-branch model, combining a convolutional neural network for spatial feature extraction and a multilayer perceptron for spectral feature extraction, enabled the dynamic and adaptive fusion of complementary information. Experimental results demonstrated that this approach significantly improved the accuracy, recall, and F1-score compared to single-modality and traditional machine learning methods, with overall accuracy increased by up to 8.0% and F1-score by 38.6%. The method supports rapid, non-destructive, and high-precision disease detection throughout different developmental stages, providing robust technical support for orchard management and postharvest quality control. Nevertheless, the model exhibited limitations in distinguishing samples with atypical symptoms or mechanical damage, indicating the need for further optimization. Future work to improve and validate model generalization and future efforts will expand the dataset to include multiple cultivars, orchards, and growing seasons. We will also explore transfer learning and domain adaptation techniques, fine-tuning with small target-site subsets and domain adversarial training to enhance cross-site robustness. Finally, we plan to release the extended dataset to facilitate reproducible cross-site evaluation and foster community comparisons. Overall, this research demonstrates the potential of multi-source data fusion combined with deep learning for intelligent fruit disease detection and provides a theoretical and technical reference for precision agriculture.

## Figures and Tables

**Figure 1 sensors-25-06763-f001:**
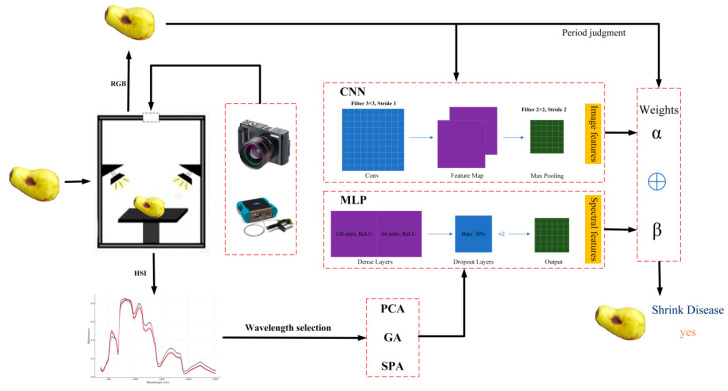
Workflow for multi-source detection of jujube shrinkage disease.

**Figure 2 sensors-25-06763-f002:**
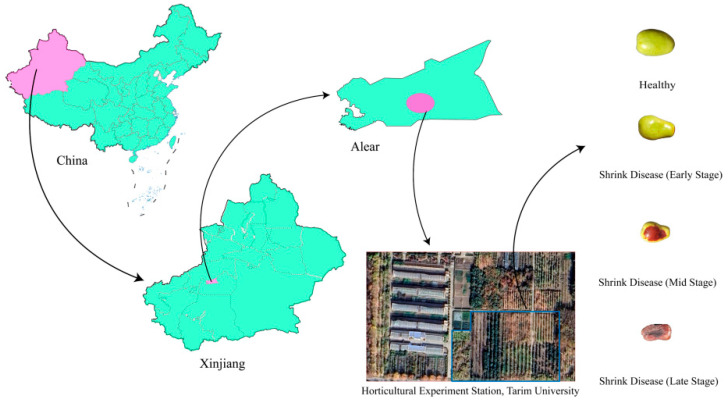
Geographical distribution of jujube shrinkage disease sampling sites.

**Figure 3 sensors-25-06763-f003:**
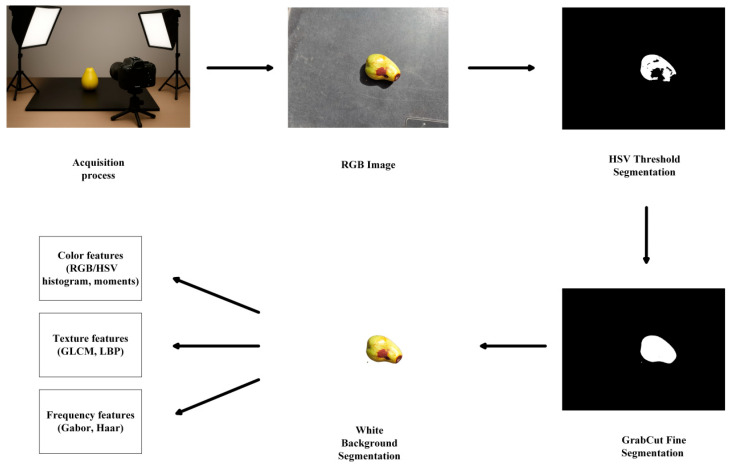
Workflow of RGB image preprocessing for jujube.

**Figure 4 sensors-25-06763-f004:**
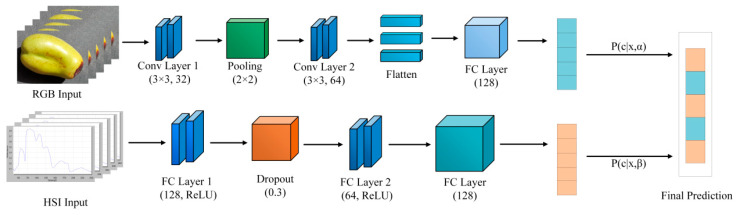
Architecture of the dual-branch multimodal fusion network.

**Figure 5 sensors-25-06763-f005:**
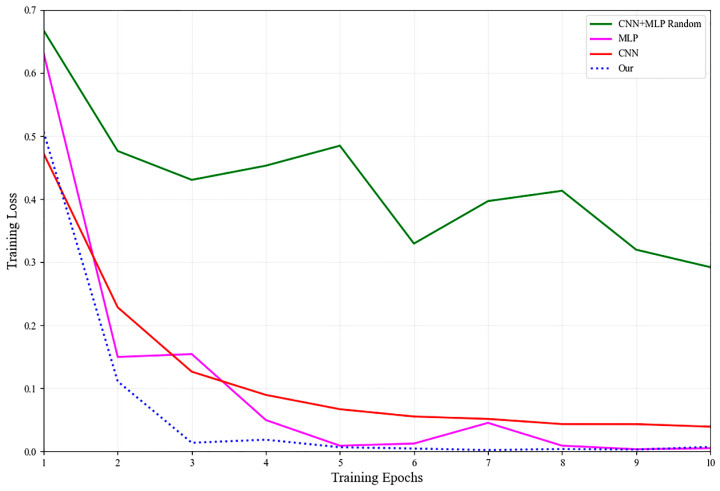
Comparison of training loss curves for different models.

**Figure 6 sensors-25-06763-f006:**
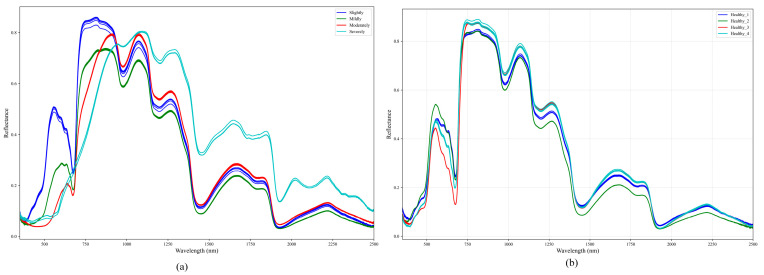
Normalized hyperspectral reflectance curves of different types of jujube. (**a**) Normalized spectral reflectance of four different grades of shriveled red dates. (**b**) Spectral reflectance of healthy red dates.

**Figure 7 sensors-25-06763-f007:**
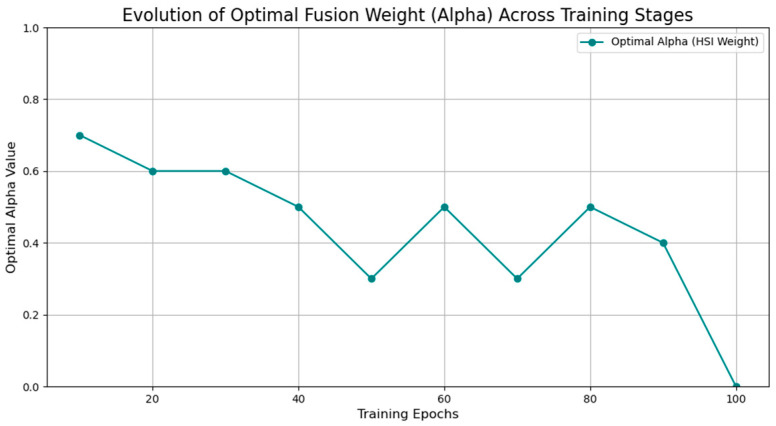
Evolution of optimal fusion weight (Alpha) across training stages.

**Figure 8 sensors-25-06763-f008:**
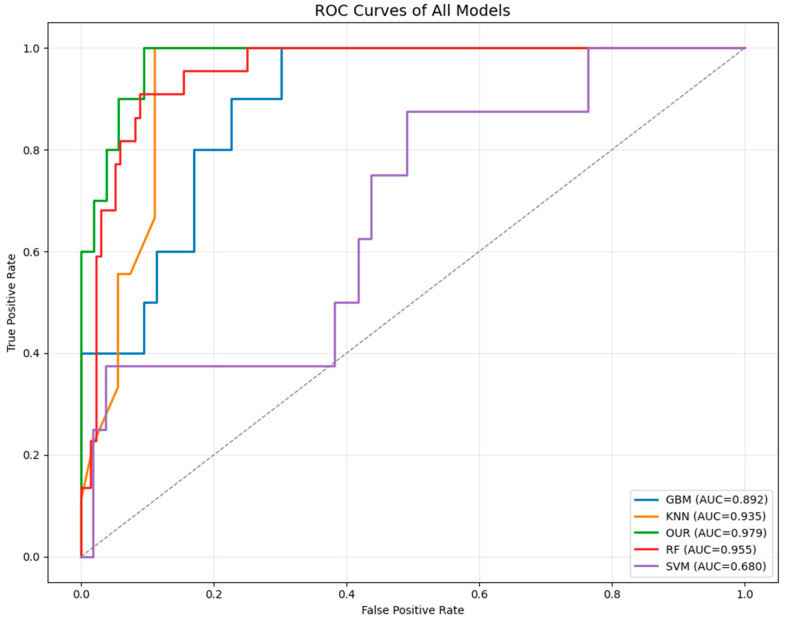
ROC curves of all models.

**Figure 9 sensors-25-06763-f009:**
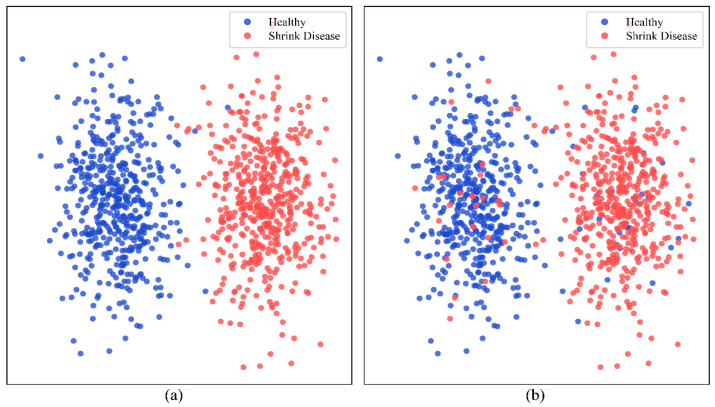
Comparison of actual and predicted distribution scatter plots of the model. (**a**) scatter plot of the actual data distribution showing healthy (blue) and shrink-diseased (red) jujube fruits; (**b**) scatter plot of the predicted classification results obtained from the model.

**Figure 10 sensors-25-06763-f010:**
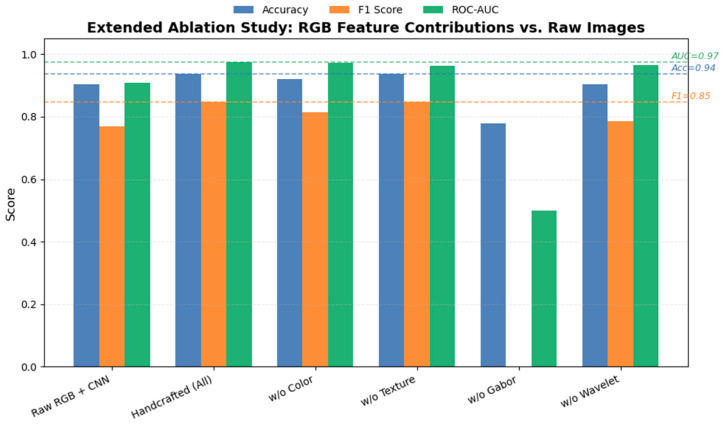
Extended ablation study on RGB feature contributions and comparison with raw RGB images.

**Figure 11 sensors-25-06763-f011:**
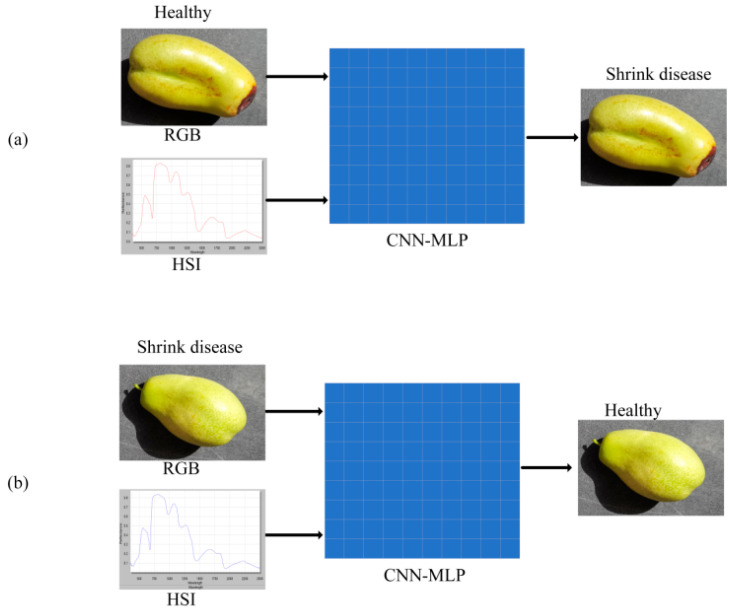
Analysis of typical misclassification cases. (**a**) false positive case—a healthy fruit misclassified as diseased due to mechanical bruising and discoloration; (**b**) false negative case—an infected fruit misclassified as healthy because early-stage infection produced no distinct visual or spectral cues.

**Table 1 sensors-25-06763-t001:** Dataset partition and sample statistics for jujube shrinkage disease.

	Samples	Raw Data
Training set	Image	257
	Jujube	1285
Validation/Test set	Image	30
	Jujube	150

**Table 2 sensors-25-06763-t002:** The most influential features extracted by different methods.

	Methods	GA (nm)	PCA (nm)	SPA (nm)
Wavelength				
1		389	535	902
2		393	536	553
3		528	537	1394
4		553	538	747
5		677	539	688
6		747	545	2267
7		778	546	1273
8		878	547	603
9		1070	553	1900
10		1075	554	483
11		1110	555	702
12		1239	596	350
13		1245	597	651
14		1579	598	1620
15		1624	684	412
16		1674	685	1075
17		1861	747	353
18		1935	883	808
19		2010	1070	673
20		2276	1075	1070

**Table 3 sensors-25-06763-t003:** Comparison of classification performance among different models.

Methods	Accuracy	F1 Score	Precision	Recall	ROC-AUC
SVM	0.794	0.381	0.333	0.444	0.680
RF	0.921	0.667	0.833	0.556	0.955
KNN	0.921	0.615	1	0.444	0.935
GBM	0.905	0.571	0.80	0.444	0.892
CNN+MLP Random	0.841	0.545	0.600	0.500	0.837
Our	0.937	0.857	0.667	0.750	0.979

**Table 4 sensors-25-06763-t004:** Performance comparison of different model structures.

Methods	Accuracy	F1 Score	Precision	Recall	ROC-AUC
MLP	0.857	0.471	0.500	0.444	0.897
CNN	0.889	0.533	0.667	0.444	0.897
CNN+KNN	0.762	0.750	0.286	0.444	0.883
SVM+MLP	0.810	0.333	0.333	0.333	0.660
CNN+MLP Random	0.841	0.545	0.600	0.500	0.837
Our	0.937	0.857	0.667	0.750	0.984

## Data Availability

The data from this study are publicly available. The dataset is available at https://github.com/2484733079/Early-detection-of-Jujube-Shrinkage-Disease-by-Multi-source-Data-on-Multi-task-Deep-Network.git (accessed on 13 October 2025).

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
