# Peer review of "Early Detection of Jujube Shrinkage Disease by Multi-Source Data on Multi-Task Deep Network"

_sensors, 2025, doi:10.3390/s25216763_

Round 1

Reviewer 1 Report

Comments and Suggestions for Authors

1、The introduction summarizes existing achievements, but it lacks a clear explanation of how this study differs from previous methods. The advantages of this research need to be further elaborated.

2、The dataset contains only 317 samples, with limited varietal diversity. It is questionable whether this is sufficient to support the generalization ability of the model.

3、Although the authors mention the use of the Adam optimizer and some hyperparameters, the description of the CNN architecture is insufficient, which reduces the reproducibility of the results.

4、The result analysis is too superficial. For example, the selection of different bands is only briefly presented without further in-depth discussion.

5、Some figures in the paper are unclear, such as Figure 5 and Figure 6. It is recommended to improve the quality of the figures.

Reviewer 2 Report

Comments and Suggestions for Authors

This manuscript addresses the topical scientific issue of "Early Detection of Jujube Fruit Damping Disease Using a Multitask Deep Neural Network," as RGB visualization and hyperspectral imaging are currently very important for recognizing plant disease symptoms at early stages. This study utilized a multi-source data fusion strategy combining RGB and HSI visualization. The author proposed a novel multilayer perceptron (CNN-MLP) neural network architecture for non-destructive and highly accurate detection of early stage plant diseases.

However, there are a number of minor comments regarding the manuscript:
1. The manuscript states that 317 fruits were collected, and 5 hyperspectral scans were obtained for each, yielding a total of 1,585 spectral samples. However, Table 1 (Section 2.2.3) indicates that the training set included 254 "images" and 1,609 "jujube" samples. This terminological confusion ("jujube" vs. "image") and numerical discrepancies make it difficult to accurately reproduce the experiment and understand the data distribution.
2. The article provides insufficient justification for the choice of key wavelengths, explaining why these specific wavelengths (553 nm, 747 nm, 1070 nm, 1075 nm) are critical for detecting desiccation. The association of these specific spectral regions with biochemical changes (e.g., chlorophyll degradation, water loss, lignification) during desiccation is only hypothetical and not supported by references to fundamental phytospectroscopic studies.
3. In the article, the authors mention that the weights stabilized around 0.5, but do not provide graphs or a detailed analysis of their training process. It would be extremely revealing to analyze how the contribution of RGB and hyperspectral features changed across different training epochs and for different sample types (e.g., early vs. late stages). 4. The authors use a "CNN+MLP Random" model with random initialization as one of the base models. However, a comparison with a deliberately unoptimized model is inappropriate for demonstrating the superiority of the proposed method. A more convincing comparison would be with other modern multimodal fusion architectures (e.g., those based on transformers or attention mechanisms).
5. The article describes in detail a complex hyperspectral data preprocessing pipeline (calibration, smoothing, feature selection). However, for RGB images, a large number of hand-crafted features (color, texture, Gabor, wavelets) are extracted. No ablation studies have been conducted to assess the contribution of these complex RGB features to the final result compared to using raw images in a CNN.
6. There is no quantitative assessment of the proportion of errors due to mechanical damage and disease, and the proportion due to the detection of truly early infections.

Reviewer 3 Report

Comments and Suggestions for Authors

In this work, a multi-source data fusion strategy was proposed for non-destructive and high-precision detection of early-stage jujube shrinkage disease, which combined RGB imaging and HSI. Moreover, CNN-MLP architecture was constructed with dynamic feature weighting, which was used to achieve effective multimodal fusion and optimize detection performance. I think the manuscript is in the scope of the journal and can be considered for the publication, and the following comments should be clarified before publish.

Comment 1: In this work, RGB imaging and HSI were combined for non-destructive and high-precision detection of early-stage jujube shrinkage diseas. However, the article did not explicitly state whether hyperspectral data is strictly aligned with RGB images. Moreover, the author needs to provide more details about the sample category distribution, such as the number of early/late stage diseased fruits.

Comment 2: The author described the sampling location, time, and physiological criteria in the article. The author constructed a dataset of 317 jujube fruits and divided it into training/validation/testing sets. However, for building deep learning models with multi-source fusion, the overall sample size is still relatively small. It is suggested that the author consider adding data from different orchards, years, and varieties to enhance the generalization and robustness of the model.

Comment 3: In this work, a multi-source data fusion strategy high-precision detection of early-stage jujube shrinkage disease. However, the author did not systematically compare the proposed method with other fusion strategies such as concatenation, attention mechanisms, weighted averaging, etc. Suggest conducting comparative experiments on different fusion methods to verify the actual contribution of the proposed dynamic fusion strategy in terms of performance improvement.

Comment 4: The experimental results demonstrate that the proposed model outperforms the comparison methods (such as MLP, CNN) in terms of accuracy and F1 score. And it also has a faster training convergence speed. However, as shown in Table 3, some machine learning methods (such as RF and KNN) also achieved high accuracy, but their F1 scores were significantly lower than the method proposed in this paper. It is suggested to further analyze whether this phenomenon is caused by an imbalanced distribution of categories. In addition, the author did not provide a ROC curve graph. Suggest the author to supplement relevant illustrations to enhance the credibility and interpretability of the experimental results.

Comments on the Quality of English Language

Can be improved.

Round 2

Reviewer 1 Report

Comments and Suggestions for Authors

The author has answered all the questions.